# Immunomorphological Patterns of Chaperone System Components in Rare Thyroid Tumors with Promise as Biomarkers for Differential Diagnosis and Providing Clues on Molecular Mechanisms of Carcinogenesis

**DOI:** 10.3390/cancers15082403

**Published:** 2023-04-21

**Authors:** Letizia Paladino, Radha Santonocito, Giuseppa Graceffa, Calogero Cipolla, Alessandro Pitruzzella, Daniela Cabibi, Francesco Cappello, Everly Conway de Macario, Alberto J. L. Macario, Fabio Bucchieri, Francesca Rappa

**Affiliations:** 1Department of Biomedicine, Neurosciences and Advanced Diagnostics (BIND), Institute of Human Anatomy and Histology, University of Palermo, 90127 Palermo, Italy; letizia.paladino@unipa.it (L.P.); radha.santonocito@unipa.it (R.S.); alessandro.pitruzzella@unipa.it (A.P.); francesco.cappello@unipa.it (F.C.);; 2Euro-Mediterranean Institute of Science and Technology (IEMEST), 90139 Palermo, Italy; econwaydemacario@som.umaryland.edu (E.C.d.M.); ajlmacario@som.umaryland.edu (A.J.L.M.); 3Department of Surgical Oncology and Oral Sciences, University of Palermo, 90127 Palermo, Italy; giuseppa.graceffa@unipa.it (G.G.); calogero.cipolla@unipa.it (C.C.); 4Consortium of Caltanissetta, University of Palermo, 93100 Caltanissetta, Italy; 5Department of Sciences for the Promotion of Health and Mother and Child Care, “G. D’Alessandro”, Pathology Institute, University of Palermo, 90127 Palermo, Italy; daniela.cabibi@unipa.it; 6Department of Microbiology and Immunology, School of Medicine, University of Maryland at Baltimore-Institute of Marine and Environmental Technology (IMET), Baltimore, MD 21202, USA

**Keywords:** thyroid cancer, Hurthle cell carcinoma, medullary carcinoma, anaplastic carcinoma, Hsp27, Hsp60, Hsp90, chaperone system

## Abstract

**Simple Summary:**

Differential diagnosis by optical microscopy on biopsies of low-incidence tumors of the same organ can be difficult, hampering patient management, particularly in the many places around the world lacking advanced facilities for cancer diagnosis beyond a histopathology laboratory. Examples of these tumors are the Hurthle cell, anaplastic, and medullary carcinomas of the thyroid. Currently, there are no specific biomarkers detectable by optical microscopy for any of them. We study the Chaperone System of these tumors by immunohistochemistry to determine if its components show distinctive levels and distribution patterns. Here we report that the molecular chaperones Hsp27, Hsp60, and Hsp90, show quantitative levels and distribution patterns different for each tumor and different from those of a benign thyroid pathology, goiter. Therefore, the reported methodology offers a promising tool to diagnose these three malignancies, and for revealing clues about the role of the three chaperones in carcinogenesis of thyroid tissue.

**Abstract:**

Hurthle cell (HC), anaplastic (AC), and medullary (MC) carcinomas are low frequency thyroid tumors that pose several challenges for physicians and pathologists due to the scarcity of cases, information, and histopathological images, especially in the many areas around the world in which sophisticated molecular and genetic diagnostic facilities are unavailable. It is, therefore, cogent to provide tools for microscopists to achieve accurate diagnosis, such as histopathological images with reliable biomarkers, which can help them to reach a differential diagnosis. We are investigating whether components of the chaperone system (CS), such as the molecular chaperones, can be considered dependable biomarkers, whose levels and distribution inside and outside cells in the tumor tissue could present a distinctive histopathological pattern for each tumor type. Here, we report data on the chaperones Hsp27, Hsp60, and Hsp90. They presented quantitative levels and distribution patterns that were different for each tumor and differed from those of a benign thyroid pathology, goiter (BG). Therefore, the reported methodology can be beneficial when the microscopist must differentiate between HC, AC, MC, and BG.

## 1. Introduction

Thyroid cancer (TC) is the most common endocrine neoplasm, with a higher prevalence in women, accounting for 3.4% of all cancers diagnosed annually worldwide [1]. Given the heterogeneity that characterizes TC, various classifications have been proposed, with the most recent categorizing TC into two main types, follicular-derived and neuroendocrine, as well as parafollicular C-cell derived (Figure 1) [1,2,3].

As we reported previously, sensitive and specific immunohistochemical identification, based on new biomarkers, is necessary, not only for the initial diagnosis, but also for a precise staging of TCs [6,7]. Among biomarkers, we focus on the members of the chaperone system (CS), which show potential, even if their role in TC is still poorly understood. In general, molecular chaperones, including heat shock proteins (Hsps), are involved in protein maturation and stabilization, playing a cytoprotective role. However, it is now well known that chaperones can cause diseases called chaperonopathies, including several types of cancer, when they are abnormal in structure, location, or quantity [8,9]. High levels of Hsps often seem to reflect the abnormal need for chaperones by tumor cells [6,7]. 

The aim of this work is to perform an immunomorphological analysis of tissue from follicular (Hurthle cell carcinoma (HC) and anaplastic carcinoma (AC) and neuroendocrine (medullary carcinoma MC) C-cell-derived tumors (Figure 1). These tumors have a low incidence, with scarce or no immunomorphological data currently available. Our intent, along with that of others [8], is to emphasize the potential involvement of Hsps in thyroid carcinoma to stimulate scientific research in the development of Hsps inhibitors as novel anticancer agents for specific subtypes of thyroid cancer, which are characterized by poor response to therapies. Therefore, the tissue expression of the molecular chaperones Hsp27, Hsp60, and Hsp90 were assessed in these carcinomas, and benign goiter (BG) samples were used as controls.

## 2. Materials and Methods

### 2.1. Tissue Samples

Formalin-fixed paraffin-embedded human thyroid tissues from benign goiter (BG), medullary carcinoma (MC), Hurthle carcinoma (HC), and anaplastic carcinoma (AC) were retrieved from the archives of the Department of Human Pathology of the University of Palermo, to perform immunohistochemical assays for Hsp27, Hsp60, and Hsp90. The samples were from patients who underwent surgical thyroidectomy at the Department of Surgical Oncology and Oral Sciences at the University of Palermo between May 2018 and March 2022. The BG group consisted of 12 tissue samples from four males and eight females, aged 25 to 70 years; the HC group consisted of samples from 12 tissue samples from four males and eight females, aged 29 to 67 years; the MC group consisted of 12 tissue samples from five males and seven females, aged 68 to 86 years; and the AC group consisted of nine tissue samples from four male and five female subjects, aged 78 to 85 years. This work was conducted as part of a project approved by the Ethics Committee of University Hospital AUOP Paolo Giaccone of Palermo (N. 05/2017 of 10 May 2017).

### 2.2. Immunohistochemistry

The immunohistochemical reactions were performed on tissue sections, 5 µm thick, obtained from paraffin blocks with a cutting microtome. The sections were dewaxed in the oven for 30 min at 60 °C before immersion in xylene for 30 min at 60 °C, and then they were rehydrated by sequential immersions in a decreasing scale of alcohols and in distilled water at 22 °C. After deparaffination, the sections were immersed for 8 min in sodium citrate buffer (pH 6) at 85 °C for antigen retrieval and then immersed for 8 min in acetone at −20 °C to prevent the detachment of the sections from the slide. After washing with PBS (phosphate buffered saline, pH 7.4) for 5 min, the sections were immunostained, using the Immunoperoxidase Secondary Detection System (Millipore, Burlington, MA, USA & Canada, cat. N° DAB-500), and the sections were then treated for 10 min with 3% hydrogen peroxide to inhibit endogenous peroxidase activity, and, after another wash with PBS at 22 °C for 5 min, they were treated by applying drops of blocking reagent (blue-colored reagent) for 5 min in a humid and enclosed container. Subsequently, the sections were rinsed with 1× Rinse Buffer for 30 s and incubated with primary antibody overnight at 4 °C. The primary antibodies used were anti-human Hsp27 (mouse monoclonal antibody, Santa Cruz Biotechnology, Dallas, TX, USA, F-4, sc-13132, dilution 1:200), anti-human Hsp60 (rabbit polyclonal antibody, Abcam, Cambridge, CB2 0AX, UK, Cat. No. ab46798, dilution 1:400), and anti-human Hsp90 (mouse monoclonal antibody, Santa Cruz Biotechnology, clone F-8, dilution 1:200). The day after, the sections were rinsed with 1× Rinse Buffer for 30 s and incubated with secondary antibody drops for 10 min at 22 °C. The sections were washed again with 1× Rinse Buffer and incubated with Streptavidin HRP drops for 10 min at 22 °C in a humid and enclosed container. After buffer wash, the sections were incubated with an appropriate volume of the freshly prepared Chromogen Reagent for 10 min in the dark inside a humid and enclosed container and after another buffer wash, incubated with the Hematoxylin Counter Stain solution for 1 min at 22 °C for nuclear counterstaining, which is shown in blue color. Finally, the slides were mounted for observation with coverslips using a permanent mounting medium (Vecta Mount, H-5000, Vector Laboratories, Inc., Burlingame, CA, USA). The slides were observed with an optical microscope (Microscope Axioscope 5/7 KMAT, Carl Zeiss, Milan, Italy) connected to a digital camera (Microscopy Camera Axiocam 208 color, Carl Zeiss, Milan, Italy) for evaluation of the immunopositivity, which is shown in brown color. Two independent observers (F.R. and F.C.) evaluated the immunostaining on two separate occasions and performed quantitative analysis to determine the percentage of immunopositivity. All observations were made at 400× magnification, and the percentage of positive cells was calculated in a high-power field (HPF) and repeated for 10 HPFs. The final value (in percentage) for each case was the arithmetic mean of the 10 values obtained, and this arithmetic mean of counts was used for statistical analysis.

### 2.3. Statistical Analysis

For comparative evaluation of the results, one-way analysis of variance (one-way ANOVA with Bonferroni post hoc multiple comparisons) was applied, using GraphPad Prism 4.0 software (GraphPad Inc., San Diego, CA, USA). Data are presented as arithmetic mean (AM) ± standard deviation (SD), and the limit of statistical significance was set at *p* ≤ 0.05.

## 3. Results

Immunopositivity for Hsp27 was observed in the cytoplasm of the epithelial cells and was 7.6 ± 3 in the benign goiter (BG) group (Figure 2A,B), 76.12 ± 7.75 in the Hurthle cell carcinoma (HC) group (Figure 2C,D), 6.6±1.9 in the medullary carcinoma (MC) group (Figure 2E,F), and 88.80 ± 4.96 in the anaplastic carcinoma (AC) group (Figure 2G,H).

Statistical analysis revealed a significant difference between HC and BG and MC (*p*-value ≤ 0.05) and a significant difference between AC and all others (*p*-value ≤ 0.05) (Figure 3).

Immunopositivity for Hsp60 was present in the cytoplasm of the epithelial cells with a granular pattern in benign goiter (BG) (Figure 4A,B), with an average of 5.50 ± 3.07, as well as with a diffuse pattern in Hurthle cell carcinoma (HC) (Figure 4C,D), medullary carcinoma (MC) (Figure 4E,F), and anaplastic carcinoma (AC) (Figure 4G,H), with an average of 73.75 ± 7.38, of 88.37 ± 3.46, and of 88.50 ± 5.97, respectively. 

Statistical analysis revealed a significant difference between the HC group and the BG and MC groups (*p*-value ≤ 0.05) and a significant difference between the AC group and the BG and MC groups (*p*-value ≤ 0.05) (Figure 5).

Immunopositivity for Hsp90 appeared diffusely in the cytoplasm and nuclei of epithelial cells with an average percentage of 9.12 ± 3.39 in benign goiter (BG) (Figure 6A,B), 58.12 ± 8.42 in Hurthle cell carcinoma (HC) (Figure 6C,D), 79.87 ± 7.47 in medullary carcinoma (MC) (Figure 6E,F), and 94.00 ± 3.08 in anaplastic carcinoma (AC) (Figure 6G,H).

Statistical analysis revealed a significant difference between HC and BG, MC, and AC (*p* ≤ 0.05) and between the AC group and all other groups (*p* ≤ 0.05) (Figure 7).

## 4. Discussion

In this work, we evaluated, by immunohistochemistry, the tissue levels of Hsp27, Hsp60, and Hsp90 in tumors of the thyroid with a low frequency and scarcely available histopathological images. These tumors are the Hurthle cell carcinoma (HC), anaplastic carcinoma (AC), and medullary carcinoma (MC), which we compared with benign goiter (BG), used as a control representing a benign disorder.

The scarcity of cases and lack of information, including histopathological images with specific biomarkers, for these tumors make the work of endocrinologists and pathologists very challenging during the initial patient identification and subsequent differential diagnosis. This is a serious problem in those parts of the world in which sophisticated diagnostic equipment for targeting molecular and genetic markers is unavailable. To contribute to remedying this situation and providing tools for those that do not have access to the most modern diagnostic laboratory settings, we are investigating, by immunohistochemistry, the components of the CS to determine whether they can be used as reliable biomarkers to aid differential diagnosis and prognostication. Furthermore, our research aims at providing information on the variations of the levels and distribution in the tumor tissue of CS components as a basis for studying their role in thyroid carcinogenesis. The chief components of the CS are the molecular chaperones, some of which are named Hsp, and we, here, report on three of them, Hsp27, Hsp60, and Hsp90. Typically, these and other chaperones are cytoprotective, but if abnormal in structure–function, quantity, and/or location, they can cause diseases, known as chaperonopathies [9,10]. Chaperones can favor carcinogenesis through various mechanisms, as shown by many experimental and clinical data [11,12,13]. For example, high tissue levels of certain Hsps are often associated with cancer progression and invasiveness, suggesting that these proteins favor, in some types of tumors, the tendency to invade surrounding tissues and to spread to distant organs [6,7,14,15]. The chaperones can favor carcinogenesis via different metabolic pathways. For instance, Hsp27 plays a protective role in cells by stabilizing denatured or aggregated proteins and returning them to their original form, functions that enable cancer cells to grow and multiply [16,17,18,19,20]. Moreover, Hsp27 can interfere with apoptotic pathways and regulate cytoskeleton dynamics [21]. High levels of Hsp27 have been observed in some types of cancer, and this has been linked to its ability to block apoptosis by interacting with several partners involved in the apoptotic pathway, ensuring cancer cell survival and growth [7,22]. 

Hsp60 is an important chaperone that maintains protein homeostasis in mitochondria [23,24,25,26,27,28]. It is upregulated in various types of tumors, acting in cancer progression by mTOR signaling pathway [7,21,25,28]. 

Hsp90 is an abundant chaperone protein expressed by all eukaryotic cells [29,30,31,32]. Hsp90 inhibits cellular apoptosis and helps nascent proteins adopt their biologically active conformations, including proteins involved in critical functions that promote tumor-cell growth, proliferation, and survival [22]. Hsp90 is often overexpressed and associated with poor prognosis in a variety of tumors because it promotes the activation of oncogenic protein kinases, e.g., JAK2/STAT3, PI3K/AKT, and MAPK, facilitating cancer progression [7,21,32,33]. This stress protein appeared to play a role in the development of medullary thyroid carcinoma and in the response to therapy [33].

In our research, in addition to performing a quantitative assessment of the three molecular chaperones mentioned above, we carried out a qualitative evaluation by observing the localization of the immunopositivity of the three molecular chaperones by mapping their distribution in the tissues because changes in localization may indicate malignancy with loss of cell differentiation [34,35,36]. We observed a higher immunopositivity of Hsp60 and Hsp90 in all groups studied, but not in the BG group, a non-malignant disorder. Immunopositivity for Hsp60, in the BG group, was observed in the cytoplasm, but it possessed a granular appearance (probably mitochondrial), whereas, in tumor cells, it was found diffusely present in the cytoplasm. From these observations, one may hypothesize that Hsp60 is involved in the carcinogenic process. Typically, in normal cells, Hsp60 is localized in the mitochondria, but, in cancer cells, it is also present in the cytosol, mostly close to the cell membrane [35,37]. Our data agree with what has been published in the literature on other types of tumors, suggesting the involvement of these Hsps in carcinogenesis and tumor progression [35]. 

Except for MC, Hsp27 showed significative differences in the HC and AC tumors, compared with BG. This result contrasts with data in the literature pertinent to follicular adenoma (FA) and carcinoma (FC) [6]. 

In a previous work, we reported an immunomorphological evaluation of papillary carcinoma (PC) [6]. In the papillary carcinoma, we found a significant increase in the levels of the three Hsps (Hsp27, Hsp60, and Hsp90) compared with BG, and we also observed their different cellular immunolocalization [6]. 

In conclusion, the immunomorphological data reported here, together with those reported earlier on FA, FC, and PC, suggest an involvement of chaperones in the mechanisms of TC carcinogenesis. This topic warrants further investigation to elucidate molecular mechanisms of chaperone participation in thyroid carcinogenesis, as well as to identify useful biomarkers and therapeutic targets for negative chaperonotherapy [7,38,39].

## 5. Conclusions

1. Despite its small size, the thyroid gland can be affected by several different tumors that are molecularly and cytologically heterogeneous, with various phenotypes and clinical presentations. This pleomorphism, even within a single tumor type, makes differential diagnosis and, consequently, adequate treatment and patient management, extremely difficult.

2. The chaperone system (CS) plays various roles in carcinogenesis, depending on the tumor type and anatomical location. Very little is known on the CS in the thyroid gland and its tumors. Quantifying and mapping at least some of the components of the CS will most likely reveal patterns distinctive of tumor type, as indicated by the results reported here and previous studies, which will help pathologists to distinguish one tumor from the other.

3. Histopathology remains the basic tenet of cancer diagnosis, and this also applies to the thyroid gland. This is still more significant if one considers that in many areas around the world there are no modern facilities with sophisticated equipment for cancer diagnosis. Thus, differential diagnosis must be achieved by examining histological sections of the tumor, using an optical microscope. Therefore, the need to standardize criteria based on histopathological images is cogent, as is the identification of markers that can provide a distinctive pattern for each tumor type. As mentioned above, quantification and mapping of selected components of the CS is a promising approach to reveal characteristic patterns for each tumor type.

4. We are characterizing the CS of the thyroid gland by immunohistochemistry, focusing on its chief components, the molecular chaperones, to determine their presence and distribution in tumor tissues and comparing different tumors and benign thyroid pathologies. Here, we present data on the molecular chaperones Hsp27, Hsp60, and Hsp90, pertaining to three thyroid tumors with a low incidence rate and limited histopathological images and information. The scarcity of images and information poses a significant problem for training pathologists and makes differential diagnosis, even by experienced microscopists, challenging.

5. The images and data on the levels and distribution patterns of the three chaperones studied indicate that they have the potential to distinguish tumors from one another. Therefore, chaperones emerge as targets for investigating molecular mechanisms of carcinogenesis characteristic of each tumor type, and for developing novel therapeutic approaches based on chaperonotherapy [8,40,41].

## Figures and Tables

**Figure 1 cancers-15-02403-f001:**
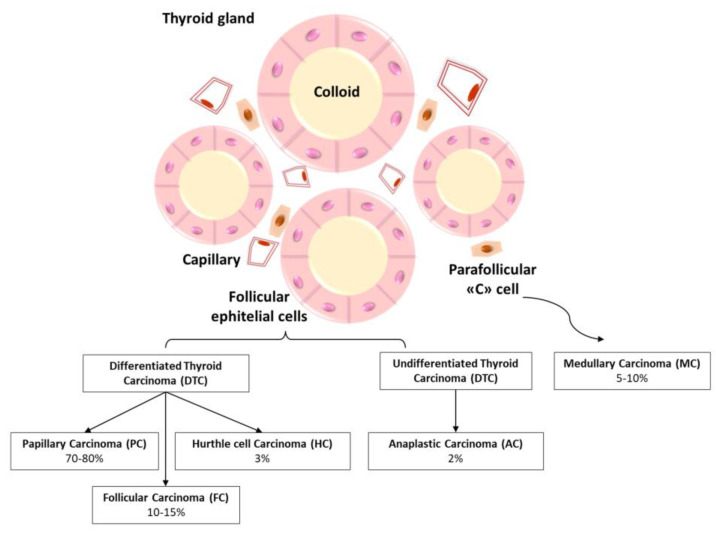
Classification of thyroid cancers (TCs). Thyroid tumors of follicular origin can be of two main subtypes: differentiated and undifferentiated. Papillary (PC) thyroid tumors are the most common type (80–85%) of well differentiated thyroid tumors and have the best prognosis. Follicular thyroid carcinoma (FC) accounts for about 10% of epithelial cancers and lacks diagnostic molecular features. Thyroidal oncocytic follicular cells can give origin to Hurthle cell carcinoma (HC), which constitute about 3% of all TC cancers. Among the undifferentiated subtype, anaplastic thyroid cancer (AC) is a rare and lethal form of TC, which probably derives from follicular cells, resulting from dedifferentiation, in patients with long-standing goiter. In addition to tumors derived from follicular cells, the thyroid gland also develops tumors from parafollicular cells, or C cells, such as the medullary thyroid carcinoma (MC), characterized by an elevated level of calcitonin [4]. Routinely, diagnosis and staging are carried out using fine-needle aspiration (FNA): cancer cells usually look different from normal cells, so the type of TC is determined by microscopic examination of thyroid cells found in nodules. In addition, molecular tests, such as those for RAS and BRAF mutations, can be used to help make a diagnosis when the result of a fine needle biopsy is uncertain [5].

**Figure 2 cancers-15-02403-f002:**
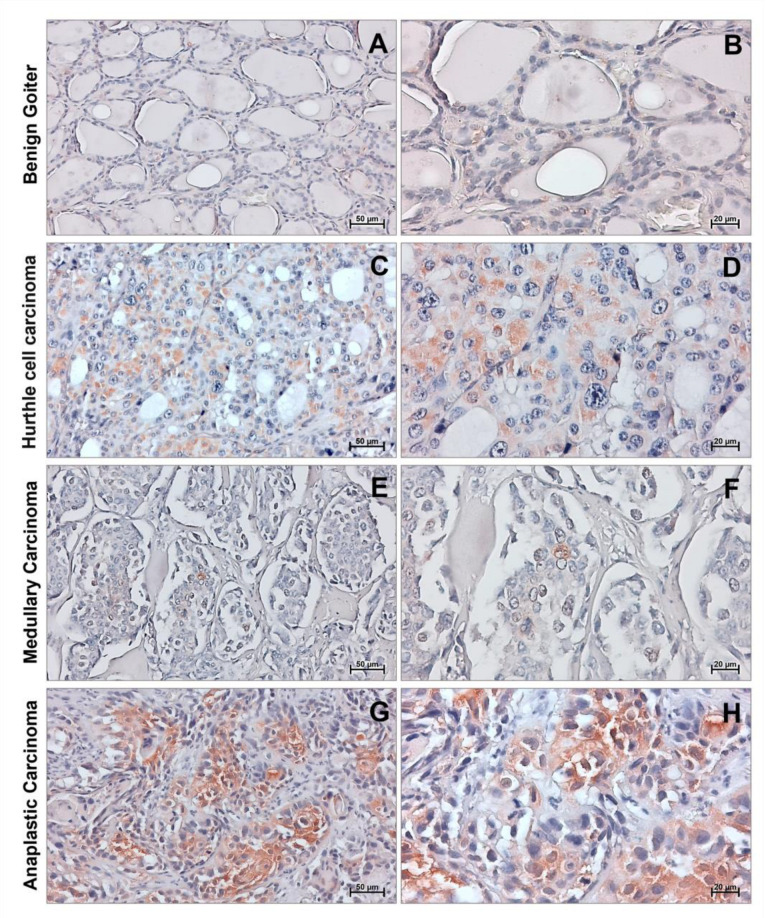
Hsp27 immunohistochemistry. Representative images of immunohistochemical results for Hsp27 in benign goiter (**A**,**B**), Hurthle cell carcinoma (**C**,**D**), medullary carcinoma (**E**,**F**) and anaplastic carcinoma (**G**,**H**). (**A**,**C**,**E**,**G**): magnification 200×, scale bar 50 µm; (**B**,**D**,**F**,**H**): magnification 400×; scale bar 20 µm.

**Figure 3 cancers-15-02403-f003:**
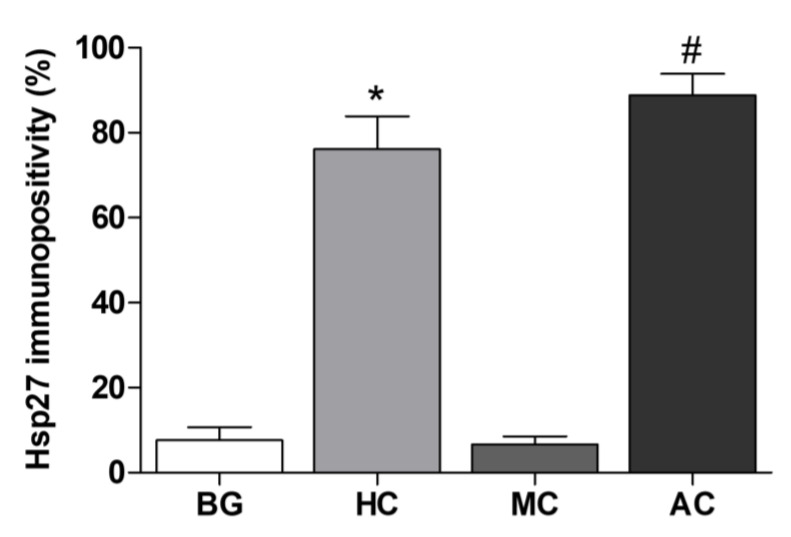
Statistical analysis of Hsp27 results. The histogram shows the results for the immunohistochemical evaluation of Hsp27 in benign goiter (BG), Hurthle cell carcinoma (HC), medullary carcinoma (MC), and anaplastic carcinoma (AC). Data are presented as arithmetic mean ± standard deviation. * *p* ≤ 0.05 vs. BG and MC # *p* ≤ 0.05 vs. BG, HC and MC.

**Figure 4 cancers-15-02403-f004:**
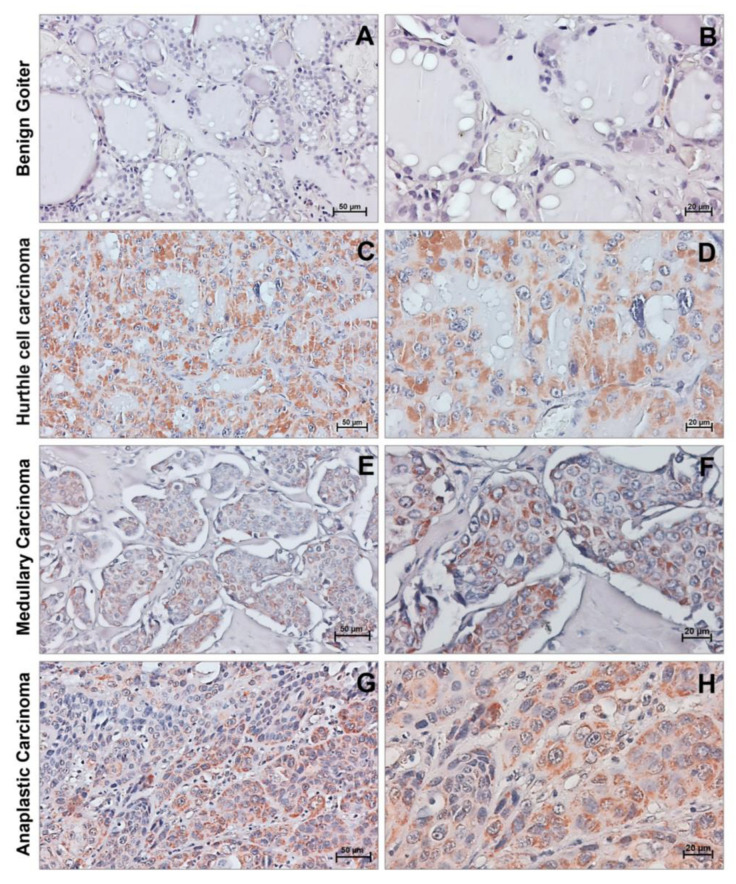
Hsp60 immunohistochemistry. Representative images of immunohistochemical results for Hsp60 in benign goiter (**A**,**B**), Hurthle cell carcinoma (**C**,**D**), medullary carcinoma (**E**,**F**), and anaplastic carcinoma (**G**,**H**). (**A**,**C**,**E**,**G**): magnification 200×, scale bar 50 µm; (**B**,**D**,**F**,**H**): magnification 400×, scale bar 20 µm.

**Figure 5 cancers-15-02403-f005:**
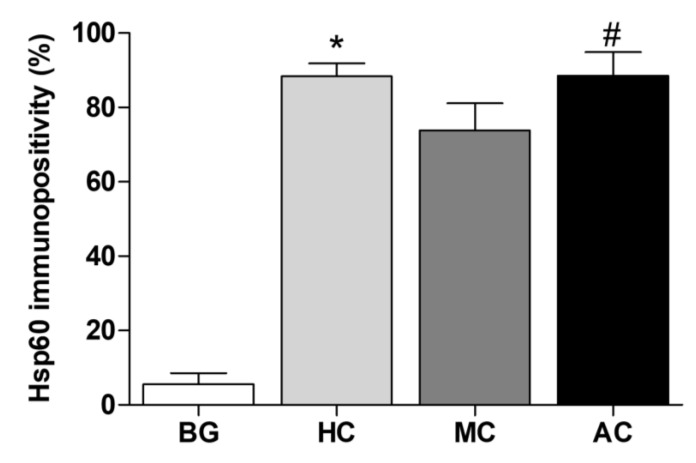
Statistical analysis of Hsp60 results. The histogram shows statistical results for the immunohistochemical evaluation of Hsp60 in benign goiter (BG), Hurthle cell carcinoma (HC), medullary carcinoma (MC), and anaplastic carcinoma (AC). Data are presented as arithmetic mean ± standard deviation. * *p* ≤ 0.05 vs. BG and MC; # *p* ≤ 0.05 vs. BG, and MC.

**Figure 6 cancers-15-02403-f006:**
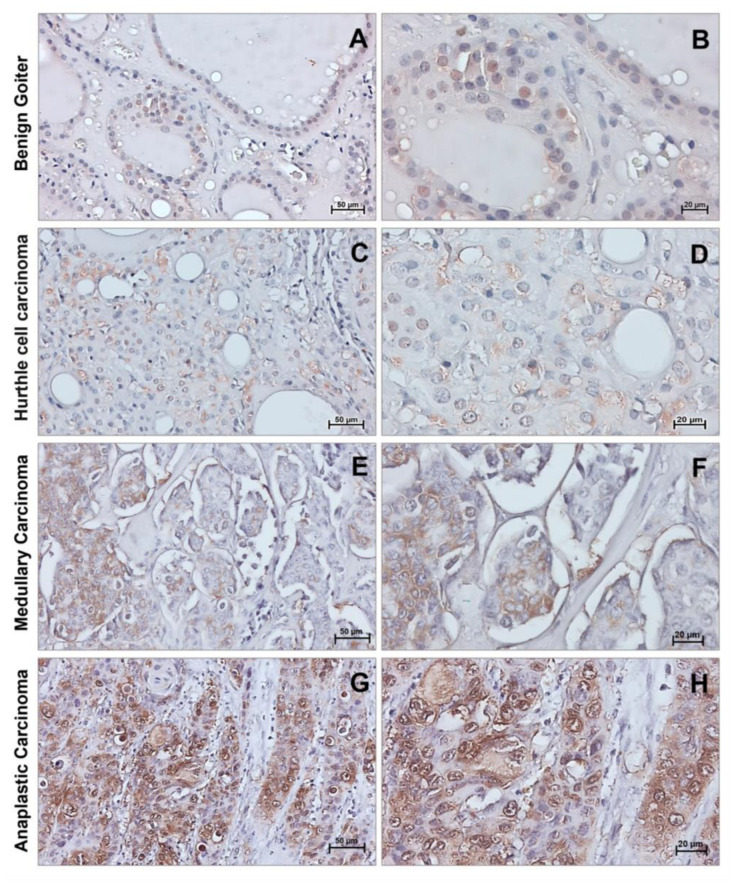
Hsp90 immunohistochemistry. Representative images of immunohistochemical results for Hsp90 in benign goiter (**A**,**B**), Hurthle cell carcinoma (**C**,**D**), medullary carcinoma (**E**,**F**), and anaplastic carcinoma (**G**,**H**). (**A**,**C**,**E**,**G**): magnification 200×, scale bar 50 µm; (**B**,**D**,**F**,**H**): magnification 400×, scale bar 20 µm.

**Figure 7 cancers-15-02403-f007:**
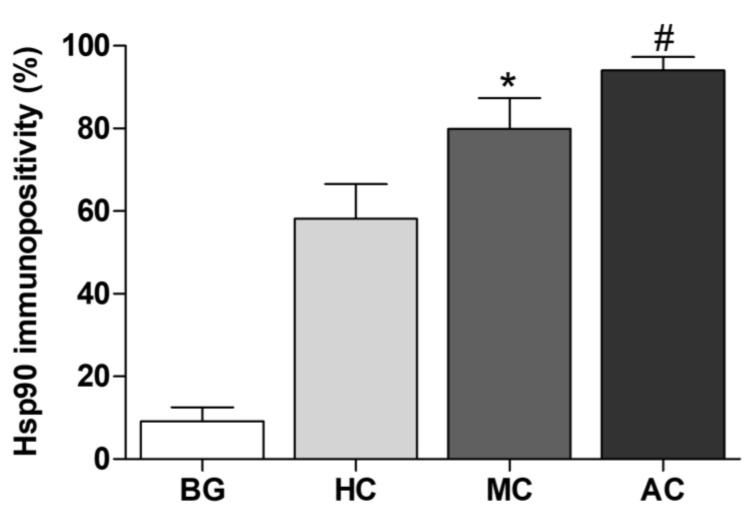
Statistical analysis of Hsp90 results. The histogram shows results for the immunohistochemical evaluation of Hsp90 in benign goiter (BG), Hurthle cell carcinoma (HC), medullary carcinoma (MC), and anaplastic carcinoma (AC). Data are presented as mean ± standard deviation. * *p* ≤ 0.05 vs. BG, MC, and AC; # *p* ≤ 0.05 vs. BG, HC, and MC.

## Data Availability

The data presented in this study are available upon request from the corresponding author. The data are not publicly available due to our preference in personal interaction with those interested in our work and data, and we are open to dialogs with colleagues who identify themselves and show genuine honest interest. We have no problems sharing data in this way.

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
