# Peer review of "Immunomorphological Patterns of Chaperone System Components in Rare Thyroid Tumors with Promise as Biomarkers for Differential Diagnosis and Providing Clues on Molecular Mechanisms of Carcinogenesis"

_cancers, 2023, doi:10.3390/cancers15082403_

Round 1

Reviewer 1 Report

The article entitled “Immunomorphological patterns of chaperone system components in rare thyroid tumors with promise as biomarkers for differential diagnosis and providing clues on molecular mechanisms of carcinogenesis” aims to evaluate by immunohistochemistry changes in the tissue levels and immunolocalization of three molecular chaperones (Hsp27, Hsp60 and Hsp90) in Hurthle cell carcinoma, medullary carcinoma and anaplastic carcinoma , using benign goiter as a pathological manifestation that most closely resembles normal thyroid parenchyma.

The topic is very interesting and opens many possibilities for the advancement of molecular research on the components of the chaperone system as biomarkers for differential diagnosis between the three

cancers studied and for the follow-up of patients. In addition, the manuscript is particularly original in that, as the authors point out several times and as I myself have seen, there are few immunomorphological data on this oncologic field in the scientific literature.

Therefore, minor improvements are needed.

1.           In lines 71-72, there is a typo: fix the parentheses;

2.           There is a typo in line 75: add a space before benign goiter;

3.           Add references on line 208: Alimardan, Zahra et al. “Heat shock proteins and cancer: The FoxM1 connection.” Biochemical pharmacology, vol. 211 115505. 15 Mar. 2023, doi:10.1016/j.bcp.2023.115505 and Tustumi, Francisco et al. “The Role of the Heat-Shock Proteins in Esophagogastric Cancer.” Cells vol. 11,17 2664. 27 Aug. 2022, doi:10.3390/cells11172664.

They may suggest interesting molecular pathways in which Hsps are involved in carcinogenesis;

4.           Given the presence of so many abbreviations in the text, I suggest that authors include a list of abbreviations at the end of the paper.

Author Response

To the Editors of the special issue

“Thyroid Cancer: New Advances from Diagnosis to Therapy” of Cancers

Subject: Manuscript ID cancers-2341511

Dear Editors,

We are submitting the revised version of the manuscript entitled “Immunomorphological patterns of chaperone system components in rare thyroid tumors with promise as biomarkers for differential diagnosis and providing clues on molecular mechanisms of carcinogenesis.”

Thanks to the Editors for having given us the opportunity to improve our manuscript. We also thank the Reviewers for their comments and suggestions, which we have followed in preparing the revision. The key revisions are in the sections mentioned in the point-by-point response to each Reviewer as they correspond, and the changes are highlighted using the "Track Changes" function in Microsoft Word. We hope that this revised version of our paper may be now satisfactory and meet the requirements for publication in the Special Issue “Thyroid Cancer: New Advances from Diagnosis to Therapy” of Cancers.

Below this message, you will find a point-by-point response to the Reviewer 1’s comments.

Best Regards,

Francesca Rappa MD, PhD

Department of Biomedicine, Neurosciences and Advanced Diagnostics (BIND), Institute of Human Anatomy and Histology, University of Palermo, Palermo, Italy. E-mail: francesca.rappa@unipa.it francyrappa@hotmail.com , Tel. (+39) 091 23865823.

Reviewer 1 Comment (R1C):

The article entitled “Immunomorphological patterns of chaperone system components in rare thyroid tumors with promise as biomarkers for differential diagnosis and providing clues on molecular mechanisms of carcinogenesis” aims to evaluate by immunohistochemistry changes in the tissue levels and immunolocalization of three molecular chaperones (Hsp27, Hsp60 and Hsp90) in Hurthle cell carcinoma, medullary carcinoma and anaplastic carcinoma , using benign goiter as a pathological manifestation that most closely resembles normal thyroid parenchyma.

The topic is very interesting and opens many possibilities for the advancement of molecular research on the components of the chaperone system as biomarkers for differential diagnosis between the three cancers studied and for the follow-up of patients. In addition, the manuscript is particularly original in that, as the authors point out several times and as I myself have seen, there are few immunomorphological data on this oncologic field in the scientific literature.

Author's Reply (AR) to the Review Report: We thank the reviewer for the positive comments.

  • Therefore, minor improvements are needed.

R1C: In lines 71-72, there is a typo: fix the parentheses;

AR: We thank the reviewer for this comment. We corrected it.

R1C: There is a typo in line 75: add a space before benign goiter;

AR: We thank the reviewer for this comment. We corrected it.

R1C: Add references on line 208: Alimardan, Zahra et al. “Heat shock proteins and cancer: The FoxM1 connection.” Biochemical pharmacology, vol. 211 115505. 15 Mar. 2023, doi:10.1016/j.bcp.2023.115505 and Tustumi, Francisco et al. “The Role of the Heat-Shock Proteins in Esophagogastric Cancer.” Cells vol. 11,17 2664. 27 Aug. 2022, doi:10.3390/cells11172664.

AR: We thank the reviewer for this comment. We have added these appropriate references.

R1C: They may suggest interesting molecular pathways in which Hsps are involved in carcinogenesis. Given the presence of so many abbreviations in the text, I suggest that authors include a list of abbreviations at the end of the paper.

AR: We thank the reviewer for this comment. We have added the list of abbreviations as requested.

Reviewer 2 Report

The author aims to determine that traditional heat shock proteins can be used as diagnostic markers for rare cancers of the thyroid. This is a good study, and I specifically looked at this proteins in the human protein database (https://www.proteinatlas.org/). The expression of proteins in thyroid cancer is up-regulation. I think it can provide a meaningful reference for the early diagnosis of thyroid cancer, but I still have some comments that need to be explained by the author. Also ignoring some work of others in this little field is surprising, changing the weight of this study from explorative to confirmatory in some aspects although intratumroal spatial distribution was never addressed before (below). Just recently, the status of the field of HSP and thyroid cancer has been summarized nicely, which was also not mentioned in current study. https://www.sciencedirect.com/science/article/abs/pii/S0303720719303788

1. Papillary thyroid carcinoma is the most common subtype, and the author does not seem to include it in the design, could you explain why?

2. I noticed that you included clinical information, is the expression of these proteins clinically relevant?

3. The description of Figure 3 needs to be consistent with the order of your pictures

4. Discussion section can add to other investigators' work on this rare subtypes, i.e. https://www.biorxiv.org/content/10.1101/2022.08.16.504041v1.full , https://www.sciencedirect.com/science/article/abs/pii/S1092913416302003
https://ar.iiarjournals.org/content/34/9/4829

5. Please keep the decimal point of the analysis results consistent

6. The picture of immunohistochemistry is not clear, please provide a clearer picture if possible

7. Methods: “independent observers (F.R. and F.C.) evaluated the immunostaining”. Some details missing- are those board-certified pathologists? Provide steps how scoring was done: quality and quantity of staining and quantity of positive cells/ out of all cells/HPF.

Author Response

To the Editors of the special issue

“Thyroid Cancer: New Advances from Diagnosis to Therapy” of Cancers

Subject: Manuscript ID cancers-2341511

Dear Editors,

We are submitting the revised version of the manuscript entitled “Immunomorphological patterns of chaperone system components in rare thyroid tumors with promise as biomarkers for differential diagnosis and providing clues on molecular mechanisms of carcinogenesis.”

Thanks to the Editors for having given us the opportunity to improve our manuscript. We also thank the Reviewers for their comments and suggestions, which we have followed in preparing the revision. The key revisions are in the sections mentioned in the point-by-point response to each Reviewer as they correspond, and the changes are highlighted using the "Track Changes" function in Microsoft Word. We hope that this revised version of our paper may be now satisfactory and meet the requirements for publication in the Special Issue “Thyroid Cancer: New Advances from Diagnosis to Therapy” of Cancers.

Below this message, you will find a point-by-point response to the Reviewer 2’s comments.

Best Regards,

Francesca Rappa MD, PhD

Department of Biomedicine, Neurosciences and Advanced Diagnostics (BIND), Institute of Human Anatomy and Histology, University of Palermo, Palermo, Italy. E-mail: francesca.rappa@unipa.it francyrappa@hotmail.com , Tel. (+39) 091 23865823.

Reviewer 2 Comment (R2C):

The author aims to determine that traditional heat shock proteins can be used as diagnostic markers for rare cancers of the thyroid. This is a good study, and I specifically looked at this proteins in the human protein database (https://www.proteinatlas.org/). The expression of proteins in thyroid cancer is up-regulation. I think it can provide a meaningful reference for the early diagnosis of thyroid cancer, but I still have some comments that need to be explained by the author. Also ignoring some work of others in this little field is surprising, changing the weight of this study from explorative to confirmatory in some aspects although intratumroal spatial distribution was never addressed before (below).  Just recently, the status of the field of HSP and thyroid cancer has been summarized nicely, which was also not mentioned in current study. https://www.sciencedirect.com/science/article/abs/pii/S0303720719303788

AR: We thank the reviewer for the positive comments. We add in the text the review suggested. We apologize if we forgot to mention the work: Lettini G, Pietrafesa M, Lepore S, Maddalena F, Crispo F, Sgambato A, Esposito F, Landriscina M. Heat shock proteins in thyroid malignancies: Potential therapeutic targets for poorly-differentiated and anaplastic tumours? Mol Cell Endocrinol. 2020 Feb 15;502:110676. doi: 10.1016/j.mce.2019.110676. We know of this study, and we had mentioned it in our review published in 2021: Paladino L, Vitale AM, Santonocito R, Pitruzzella A, Cipolla C, Graceffa G, Bucchieri F, Conway de Macario E, Macario AJL, Rappa F. Molecular Chaperones and Thyroid Cancer. Int J Mol Sci. 2021 Apr 18;22(8):4196. doi:10.3390/ijms22084196.

R2C: Papillary thyroid carcinoma is the most common subtype, and the author does not seem to include it in the design, could you explain why?

AR: In this study, we were mainly concerned with evaluating Hsp27, Hsp60, and Hsp90 in tumors less frequent than papillary carcinoma. We had already presented the immunohistochemical data on papillary carcinoma for the same proteins in an article published in 2019 in the International Journal of Molecular Sciences (Caruso Bavisotto C, Cipolla C, Graceffa G, Barone R, Bucchieri F, Bulone D, Cabibi D, Campanella C, Marino Gammazza A, Pitruzzella A, Porcasi R, San Biagio PL, Tomasello G, Conway de Macario E, Macario AJL, Cappello F, Rappa F. Immunomorphological Pattern of Molecular Chaperones in Normal and Pathological Thyroid Tissues and Circulating Exosomes: Potential Use in Clinics. Int J Mol Sci. 2019 Sep 11;20(18):4496. doi: 10.3390/ijms20184496).

R2C: I noticed that you included clinical information, is the expression of these proteins clinically relevant?

AR: We believe that the expression of these proteins is certainly remarkably relevant as already demonstrated in the literature. Our work provides immunomorphological data which on the one hand expand the information already present in the literature and on the other lays the foundations for subsequent biomolecular studies with clinical implications.

R2C: The description of Figure 3 needs to be consistent with the order of your pictures.

AR: We thank the reviewer for this comment. We corrected the legends of the figures.

R2C: Discussion section can add to other investigators' work on this rare subtypes, i.e. https://www.biorxiv.org/content/10.1101/2022.08.16.504041v1.full , https://www.sciencedirect.com/science/article/abs/pii/S1092913416302003 https://ar.iiarjournals.org/content/34/9/4829

AR: We thank the reviewer for this comment, and we added references where appropriate.

R2C: Please keep the decimal point of the analysis results consistent.

AR: We are not sure what is meant by this question. We checked the decimal point everywhere and found it to be OK. If this issue still requires our attention, please, let us know and we will proceed as required.

R2C: The picture of immunohistochemistry is not clear, please provide a clearer picture if possible.

AR: We thank the reviewer for this comment. We have added more descriptive details in the text.

R2C: Methods: “independent observers (F.R. and F.C.) evaluated the immunostaining”. Some details missing- are those board-certified pathologists? Provide steps how scoring was done: quality and quantity of staining and quantity of positive cells/ out of all cells/HPF.

AR: The authors Francesca Rappa and Francesco Cappello are two Medical Doctor and Pathologist specialized in Pathological Anatomy.

The quantitative evaluation performed by us was performed on 10 high magnification fields per case. The value attributed is the percentage of positive cells for each field. The final value used for the statistical evaluation is the mean of the 10 values.